# Leveraging Tumor Heterogeneity: Heterogeneous Graph Representation Learning for Cancer Survival Prediction in Whole Slide Images

**Junxian Wu[1,2]***   **Xinyi Ke[3]***   **Xiaoming Jiang[4]***   **Huanwen Wu[3]†**   **Youyong Kong[2,5]†**
**Lizhi Shao[1]†**

[1] School of Internet, Anhui University

[2] Jiangsu Provincial Joint International Research Laboratory of Medical Information Processing,
School of Computer Science and Engineering, Southeast University

[3] Department of Pathology, State Key Laboratory of Complex Severe and Rare Disease,
Molecular Pathology Research Center, Peking Union Medical College Hospital,
Chinese Academy of Medical Sciences and Peking Union Medical College, Beijing, China

[4] School of Bioinformatics, Chongqing University of Post and Telecommunications

[5] Key Laboratory of New Generation Artificial Intelligence Technology and Its
Interdisciplinary Applications (Southeast University), Ministry of Education, China

```
{junxianwu, kongyouyong}@seu.edu.cn
    xinyike0321@student.pumc.edu.cn
        jiangxm@cqupt.edu.cn
     wuhuanwen@xhyy.pumc.edu.cn
           24115@ahu.edu.cn
```

## Abstract

Survival prediction is a significant challenge in cancer management. Tumor micro-environment is a highly sophisticated ecosystem consisting of cancer cells, immune cells, endothelial cells, fibroblasts, nerves and extracellular matrix. The intratumor heterogeneity and the interaction across multiple tissue types profoundly impact the prognosis. However, current methods often neglect the fact that the contribution to prognosis differs with tissue types. In this paper, we propose ProtoSurv, a novel heterogeneous graph model for WSI survival prediction. The learning process of ProtoSurv is not only driven by data but also incorporates pathological domain knowledge, including the awareness of tissue heterogeneity, the emphasis on prior knowledge of prognostic-related tissues, and the depiction of spatial interaction across multiple tissues. We validate ProtoSurv across five different cancer types from TCGA (*i.e.*, BRCA, LGG, LUAD, COAD and PAAD), and demonstrate the superiority of our method over the state-of-the-art methods. Source code is available at https://github.com/wjx-error/ProtoSurv.

## 1 Introduction

Pathological images are considered the gold standard for cancer diagnosis and provide rich prognostic information, such as the tumor differentiation and lymphovascular infiltration. Traditional manual evaluation by pathologists is subject to inter-observer inconsistency and lacks accuracy in the risk stratification for patients. The advent of whole-slide imaging allows the entire slide to be digitalized at high resolution, enabling the standardized and automated analysis of Whole Slide Images (WSIs)

---

*Equal contribution.

†Corresponding author.

38th Conference on Neural Information Processing Systems (NeurIPS 2024).

with deep learning methods. Deep learning has been applied to a series of medical tasks, including tumor segmentation, grading, subtyping and the prediction of molecular alternations and clinical outcomes.

The gigapixel WSIs encompass detailed cellular-level information, but this leads to extremely high memory usage. To make the memory usage acceptable for analyzing WSIs, most current works are based on the Multiple Instance Learning (MIL) framework [1, 2, 3, 4]. In MIL, WSIs are divided into multiple instances that are encoded separately, and then aggregated to obtain bag-level (slide-level) representations for downstream tasks. However, these methods based on MIL do not emphasize the contextual information among instances within the WSI, leading to a loss of structural information across tissues. Therefore, these methods struggle to achieve good performance on prognostic prediction tasks that require a comprehensive understanding of the local morphology and overall structure of the tumor microenvironment (TME) [5]. In recent years, graph neural network (GNN) has shown tremendous potential in prognostic prediction [5, 6, 7, 8], which can learn spatial interaction across tissues, enabling the deciphering of the tumor ecosystem based on the proximity of tumor cells to other TME components [5, 7, 8].

Apart from the spatial relationships, the high-resolution WSIs contain rich information about tissue heterogeneity within a tumor [9, 10], which also has significant prognostic value. The tissue categories in pathology slides include tumor, stroma, immune infiltration, nerves, necrosis, etc., which together form the tumor microenvironment, but each varies in importance to cancer prognosis [11, 12, 13]. Considering the presence of intratumoral tissue heterogeneity and understanding the specific characteristics of crucial tissue types can enhance the medical interpretability and optimize the graph model theoretically. (i) From the medical perspective: A large amount of histological studies have established the prognostic value of certain tissue types, such as the tumor, immune infiltration, stroma, and necrosis[9, 10, 11, 12, 13]. Leveraging prior knowledge about intratumoral tissue heterogeneity can guide the model to focus on tissues highly relevant to survival prediction, aligning more closely with medical consensus and enhancing the model's interpretability. (ii) From the model design perspective: Commonly used GCN-like models are based on the homogeneity assumption, and it has been demonstrated that they do not perform well on heterogeneous graphs [14, 15]. Thus, the objective presence of heterogeneity among patches within WSIs could be affecting the performance of homogeneous graph-based WSI analysis models.

Therefore, we propose ProtoSurv, a novel heterogeneous graph model for cancer prognosis prediction. The heterogeneous graph introduces a "tissue category" attribute to each node to differentiate prognosis-related tissues in the WSIs. The selection of tissue categories is according to tissues clinically proven to be highly related to prognosis, which introduces clinical prior knowledge into the model. We incorporate the concept of prototype learning from advanced heterogeneous graph solutions [16, 17], decoupling the model into the Structure View and the Histology View. The Structure View (SV) utilizes the neighbor message-passing mechanism of GNN to simulate the operation of pathologists observing at multiple magnifications. The Histology View (HV) extract prototypes from global features under the guidance of pathological priors related to prognosis. Subsequently, under the guidance of prototypes extracted by HV, the model aggregates regions of interest from the context-aware multi-hop neighborhood information from SV. We extensively evaluated our method on five TCGA public benchmark datasets and compared it to various state-of-the-art survival prediction methods. The survival prediction results of our approach significantly outperform the competitors.

We summarize our main contributions as follows:

1. **Domain knowledge awareness:** To holistically depict the morphological features and spatial interaction across multiple tissue types within tumors, we proposed ProtoSurv, which decipher intratumoral tissue heterogeneity using a heterogeneous graph and incorporates prior knowledge of prognostic tissue types into the prediction process.

2. **Validation on multi-cancer datasets:** We conducted comprehensive evaluations on five public benchmark datasets. ProtoSurv demonstrates robust survival prediction performance across multi-cancer datasets.

## 2 Related Work

### 2.1 Weakly Supervised Learning Survival Prediction in WSIs

Manual annotations of WSIs demand enormous effort and domain knowledge from highly skilled pathologists. Consequently, only slide-level labels are commonly available, while pixel- or region-level annotations are seldom present. Hence, WSI tasks are frequently regarded as weakly supervised learning problems. In recent years, Convolutional Neural Network (CNN)-based and MIL-based weakly supervised learning approaches have been proposed for survival analysis in WSIs [18, 19, 20, 5, 8]. Mobadersany et al. [18] proposed an end-to-end CNNs method for processing manually annotated ROIs. Zhu et al. [19] used K-means to cluster patches and employed the clustering results as inputs into the CNN. Chen et al. [5] employed the MIL-based GCN method to model the topological relationships between patches, achieving context-awareness. Di et al. [8] introduced hypergraphs into survival prediction and designed strategies to overcome the limitations of sampling scales in constructing large hypergraph models.

### 2.2 Graph-based Approaches in WSIs

Graph-based MIL approaches, which model the interactions between instances via graphs, have been widely utilized in WSI analysis, solving problems such as cancer classification[6, 21, 22], cancer grading [23, 24, 25], and survival analysis [5, 26]. Chen et al. [5] used GCN [27] in the information propagation process to achieve context-awareness. Zheng et al. [22] applied graph transformer network [28] to the information propagation stage in MIL. Lee et al. [26] proposed a method to aggregate similar patches into a superpatch according to cosine similarity, and used GAT for message passing between superpatches. Despite the significant success of graph-based methods in various tasks, current approaches do not account for the inherent heterogeneity between patches and overlook the guidance of clinical prior knowledge from pathology. Chan et al. [23] highlighted the importance of heterogeneous patch categories and subsequently introduced a heterogeneous graph model called HEAT. HEAT employs HoverNet to classify each patch based on the types of cells within it and introduces heterogeneous edges to model the relationships between heterogeneous nodes.

### 2.3 Heterogeneous Graph Neural Networks

GNN models such as GCN [27] and GAT [29] have performed profoundly well on several WSI analysis tasks [5, 26]. However, GCN-like models have an inherent assumption of homophily on graphs, and many studies have highlighted their poor performance on heterogeneous graphs [14, 15]. Many works have attempted to address the issue of message passing among heterogeneous nodes within heterogeneous graphs. Early works attempted to solve the problem by aggregating information from multi-hop neighbors or by constructing auxiliary graph structures based on node and structure features [30, 31]. Recently, aggregating global information has emerged as a new direction for addressing challenges in heterogeneous graphs. Li et al. [32] found that capturing more global information substantially improves the model's performance. Some studies introduced class prototypes into models, extracting global information based on node categories [16, 17, 33]. Our model integrates the concepts from state-of-the-art heterogeneous graph methods, constructing a heterogeneous graph, capturing global information based on pathological prior categories. It incorporates the pathological priors of prognosis-relevant tissues into the model, optimizing both the model structure and pathological interpretability.

## 3 Preliminaries

**Heterogeneous Graph.** A heterogeneous graph is defined by a graph $\mathcal{G} = (\mathcal{V}, \mathcal{E})$, where $\mathcal{V} = \{v_i\}_{i=1}^N$ is the set of nodes which contains $N$ nodes and $\mathcal{E} \subseteq \mathcal{V} \times \mathcal{V}$ is the set of edges on the graph. $A$ denotes the adjacency matrix of the graph, where $A_{ij}$ represents the edge $e_{i,j}$ between nodes $v_i$ and $v_j$. $X$ represents the feature matrix of nodes, where $x_i$ is the features of node $i$. $C$ denotes the node class labels, and $c_i$ is the label for node $v_i$. Every node $v_i$ within the heterogeneous graph has a category label $c_i$ and a $d$-dimensional feature $x_i \in \mathcal{X}$, where $\mathcal{X}$ is the embedding space of node features.

**Survival Prediction in Whole Slide Images.** Given a WSI, we wish to predict the survival risk $Y$ with a prediction model. We use the clinical survival time of patients as labels, and measure

prediction performance by the relative ranking of predicted survival risk against the actual survival times of the patients.

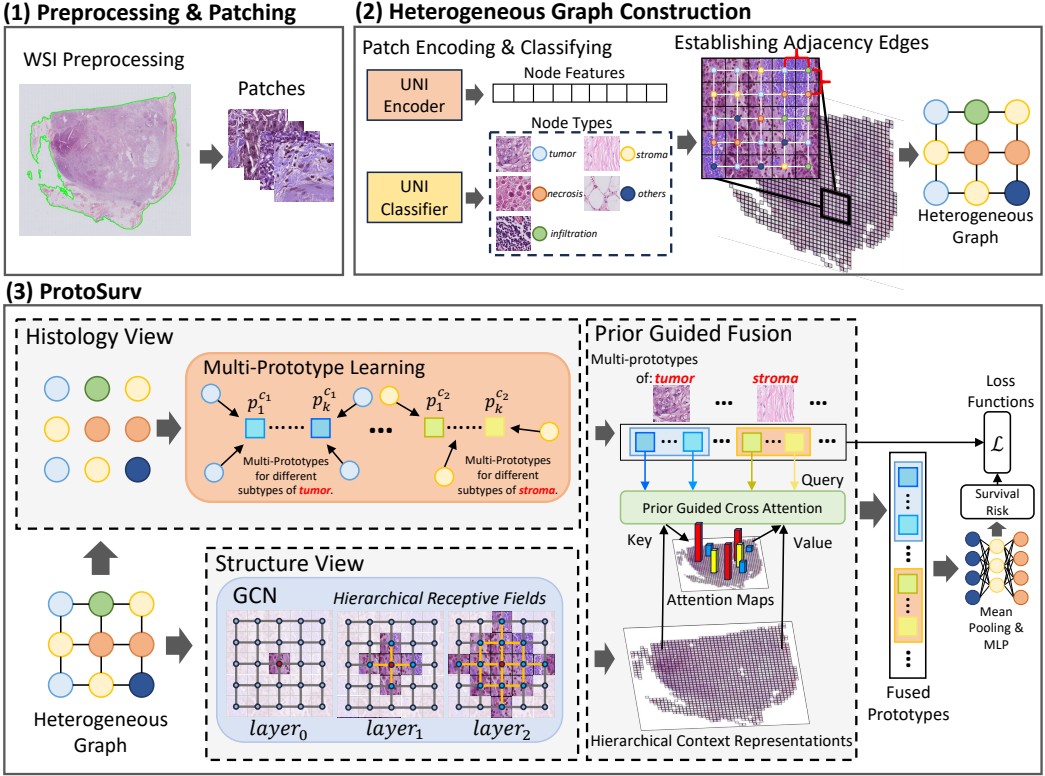

Figure 1: An overview of the ProtoSurv architecture. We use the encoder and classifier pretrained from UNI to obtain feature representations and node types for each patch. Edges are created between spatially adjacent patches to obtain a heterogeneous graph. ProtoSurv decouples the graph into two views: Structure View and Histology View. The Structure View utilizes GCN to provide multi-hop neighborhood information. The Histology View breaks the edge constraints and leverages tissue heterogeneity information to learn global multi-prototype representations for each tissue category. We use additional loss functions to regularize the multi-prototypes.

# 4 Method

## 4.1 Construction of Heterogeneous Graph

We first introduce our methodology of modeling the WSI with a heterogeneous graph. We follow Lu et al. [34] to segment foreground regions and split WSIs into non-overlapping patches of size $256 \times 256$ by sliding window strategy at 20× magnification. We use the UNI model [35] as a patch encoder to obtain patch embeddings for each patch. Pathology foundation models such as UNI are trained on large-scale pathology datasets and have been validated to represent pathological patches effectively. To incorporate pathological prior knowledge into our model, we focus on tissue regions that are recognized by the current consensus as highly relevant to cancer prognosis: tumor [11], tumor stroma [12], immune infiltration [13], and necrosis [36]. Therefore, we further fine-tune UNI to obtain our patch classifier, which classifies each patch into one of the five categories (tumor, stroma, immune infiltration, necrosis, others). The classifier training and usage details can be found in section 5.2. Each patch is treated as a node in the graph, with the representation encoded by the UNI as the node's features, and the five histological categories as the node's labels. We exploit the graph structure to simulate the topological relationship of patches within the WSI. For each node $v \in V$, we use the k-nearest neighbor algorithm to find k nodes closest to the given node in Euclidean space (k=4), and connect edges between node $v$ and its neighboring nodes. All edges have the same

weight. As a result, we obtain a heterogeneous graph $G$. The edges of the graph model the topological relationships between patches.

## 4.2 ProtoSurv

To capture scattered but significant information from prognostically relevant tissues while preserving multi-scale context perspective, inspired by Dong et al. [16], we propose ProtoSurv, which decouples the Structure View and the Histology View from the graph, allowing separate learning of hierarchical features and global features of prognosis-related tissue categories.

**Structure View.** We use GNN to learn the structure representation of WSIs. After the i-th layer of message passing, the i-th output node representations reflect the receptive field within i-hop neighbors centered around the node. We retain all output features from each GCN layer to enable the Structure View to leverage multi-hop neighborhood information. We concatenate the features from each GCN layer along the feature dimension and use a MLP to learn representations from multi-hop neighborhood information, obtaining the final feature $H$ of the Structure View.

$$h^l = \text{GCN}^l(h^{l-1}, A) \in \mathbb{R}^{N \times d} \quad \text{where} \quad h^0 = X, \tag{1}$$

$$H' = \text{concat}\left[h^1, h^2, \ldots, h^L\right] \in \mathbb{R}^{N \times Ld}, \tag{2}$$

$$H = MLP(H') \in \mathbb{R}^{N \times d_h}, \tag{3}$$

where $X$ represents the initial feature matrix of nodes, $d$ is the hidden dimension of each GCN layer, $L$ is the number of GCN layers, $N$ is the number of nodes, and $d_h$ is the hidden dimension of the Structure View.

**Histology View.** In order to fully exploit representations of tissue categories highly related to prognosis within the WSI, Histology View learns prototype representations for each category from global nodes. These prototypes extract feature information within the feature space of certain categories irrespective of their topological edges.

In clinical practice, heterogeneity exists even within the same tissue category[37, 38]. Take stroma as an example, Xu et al. [38] highlighted: *"The tumor stroma is highly dynamic, heterogeneous and commonly tumor-type specific, ... "* Therefore, using a single prototype to represent a whole tissue category is insufficient and often inaccurate. We introduce multiple prototypes for each tissue category, allowing them to capture different phenotypes within tissue feature distributions. For a detailed illustration of multiple subtypes of specific tissue categories, refer to appendix F.

In Histology View, all node features with category $c$ are first averaged to obtain the initial prototype $p_{init}^c$ of category $c$ with the hidden dimension $d$. For a visual demonstration of the multi-prototype extraction process, please refer to appendix B.

$$p_{init}^c = MEAN(X_c) \in \mathbb{R}^{1 \times d}, \tag{4}$$

Then, learnable parameters $Z = \{z_1, z_2, \ldots, z_k\}$ are used to shift the initial prototype to multi-prototypes, we obtain $K$ distinct prototypes for category $c$ that focus on different subtypes of the specific tissue. The learnable parameters $Z$ are initialized by Xavier initialization [39].

$$P_{prior}^c = \{p_c^1, p_c^2, \ldots, p_c^k\} = \{p_{init}^c + z_1, p_{init}^c + z_2, \ldots, p_{init}^c + z_k\} \in \mathbb{R}^{K \times d}, \tag{5}$$

We further employ the cross-attention mechanism to aggregate scattered global node information into the prototypes, thereby updating the multi-prototypes. Considering the potential omissions from pseudo-labeled tissue categories from the classifier and the need to focus on other relevant tissues, we extract information from global nodes, not just nodes with category $c$, to update the prototypes.

$$P_c = \text{Softmax}\left(\frac{\left[P_{prior}^c W^Q\right]\left[XW^K\right]^{\text{T}}}{\sqrt{d}}\right)(XW^V) \in \mathbb{R}^{K \times d}, \tag{6}$$

where $d$ is the hidden dimension, $W^Q$, $W^K$ and $W^V$ are query, key and value transformation matrices respectively. Finally, we obtain multiple prototypes $P$ for all categories.

$$P = \{P_1, P_2, \ldots, P_c\} \in \mathbb{R}^{CK \times d}, \tag{7}$$

The learning of prototypes can be viewed as intentionally adding edges based on pathological priors. Guided by prior knowledge, it allows global messages to pass from nodes to the interested histological category prototypes.

**Prior Guided Fusion&Pooling.** Once we have all the category prototypes $P$ from the Histology View and hierarchical context embeddings $H$ from the Structure View, we can select regions of interest within multi-hop neighborhoods from Structure View under the guidance of category prototype priors. This process can also be viewed as pooling guided by prior knowledge of the tissue categories of interest. We use the cross-attention mechanism to aggregate Structure View features and pathological multi-prototypes.

$$P_{fusion} = \text{Softmax}\left(\frac{[PW^Q][HW^K]^{\text{T}}}{\sqrt{d}}\right)(HW^V) \in \mathbb{R}^{CK \times d}, \tag{8}$$

where $W^Q$, $W^K$ and $W^V$ are query, key and value transformation matrices respectively. Guided by histopathological prior knowledge, we select regions of interest at multiple scales through cross-attention, obtaining $K$ representations for each of $C$ tissue categories, resulting in $CK$ categorized representations in total. Finally, we use average pooling to derive the WSI representation $h_{slide}$, which is used for predicting survival risk $Y$.

$$h_{slide} = MEAN(P) \in \mathbb{R}^{1 \times d}, \tag{9}$$
$$Y = g(h_{slide}), \tag{10}$$

where $g$ is a survival prediction head, which is a Multi-Layer Perceptron (MLP) used to predict survival risk $Y$ from WSI-level feature $h_{slide}$.

**Loss Functions.** To fully exploit the capabilities of the multiple prototypes of each tissue category, besides the commonly used Cox regression loss [40] $\mathcal{L}_{cox}$ for survival prediction, we introduce compatibility loss and orthogonality loss to regularize training. The compatibility loss regularizes by encouraging the prototypes and representations of a specific category to agree. Meanwhile, the orthogonality loss encourages different prototypes of the same category to be distinct.

Following Dong et al. [16], we modify the compatibility loss function from Snell et al. [41] to handle multiple prototypes within each category. Within the compatibility loss, we treat node features that belong to the same category of the prototypes as positive samples, and those that belong to different categories as negative samples, and use the negative log-likelihood loss to constrain their relationships. First, we map all prototype representations back to the latent space of the node features through a nonlinear mapping layer. Then, we calculate positive similarity scores of positive samples between prototypes and node features belonging to category $c$, as well as similarities of negative samples with features not belonging to the category. Finally, we aggregate the similarity scores of multiple prototypes into a single score for a specific category, and adopt the negative log-likelihood loss to regulate the relationship between the prototype and node features of the same category. The compatibility loss is computed as follows:

$$s_i^c = \underset{k \in K}{MEAN}\Big(\gamma(X^c, f(p_k^c))\Big), \tag{11}$$

$$\mathcal{L}_{comp} = \frac{1}{CN} \sum_{i \in N} \left[ s_i^{c_i} + \log \sum_{c' \neq c} \exp\left(-s_i^{c'}\right) \right], \tag{12}$$

where $N$ is the number of nodes, $C$ is the number of categories, $c_i$ is the category of $i^{th}$ node, $p_k^c$ is the $k^{th}$ prototype of category $c$, $X^c$ is the initial node features with category $c$, $f$ is an MLP and $\gamma$ is a similarity function. To exploit multi-prototypes, we regularize them so that they are distinct from each other, and focus on different representations of certain tissue categories. We employ orthogonality loss [42] to enforce that the prototypes are orthogonal to each other to achieve this purpose,

$$\mathcal{L}_{ortho} = \left\| \frac{(P^c)^T P^c}{\|(P^c)^T P^c\|_F} - \frac{I_d}{\sqrt{d}} \right\|_F, \tag{13}$$

where $\| \cdot \|_F$ indicates the Frobenius norm, and $I_d$ is an identity matrix. With the $\alpha$ and $\beta$ as tuning hyperparameters, the full loss function used for training is:

$$\mathcal{L} = \mathcal{L}_{cox} + \alpha\mathcal{L}_{comp} + \beta\mathcal{L}_{ortho}. \tag{14}$$

# 5 Experiments

## 5.1 Datasets

The cancer types of WSIs we use are: Breast Invasive Carcinoma (BRCA) (1064 cases), Lower Grade Glioma (LGG) (841 cases), Lung Adenocarcinoma (LUAD) (512 cases), Colon Adenocarcinoma (COAD) (441 cases), Pancreatic Adenocarcinoma (PAAD) (208 cases). All cancer types of WSIs are from The Cancer Genome Atlas (TCGA) repository.[1] We choose these cancer types for training and evaluation using the following criteria: 1) overall survival available, and 2) balanced distribution of uncensored-to-censored patients. During dataset construction, we only preserve formalin-fixed paraffin-embedded hematoxylin and eosin (H&E) slides, given the morphological alterations found in frozen sections.

## 5.2 Implementation Details

**Patch Extraction and Encoding.** First, we use OTSU to extract foreground tissue regions. Then we extract a series of non-overlapping patches at $20\times$ magnification with size $256 \times 256$ which contain more than $50\%$ foreground tissue. All patches are encoded by UNI [35] into 1024-dimensional vectors, and the encoder does not perform data augmentation during inference.

**Patch Classifier Training.** We use a small amount of proprietary annotated patches from the TCGA dataset and several public tissue classification datasets [43, 44, 45, 46] to train our classifier. The classifier is based on UNI [35]. We used the 1024-dimensional vector obtained from UNI encoding and added a classification head to classify the patches into 12 categories. We trained the classifier based on the pre-trained weights of UNI, fine-tuning without freezing. Our 12-class classifier achieved an accuracy of 92.5% on the validation set. For details on the 12 categories, refer to section 5.5.

**Network Hyper-Parameter.** For Histology View, the number of prototypes $K$ per category is set to 8, and the dimension of each prototype is set to 768. Based on pathological prior knowledge, we divided the nodes into five categories. For a detailed discussion on the tissue category selection, refer to section 5.5. For Structure View, the number of GCN layers $L$ is set to 4, the hidden dimension $d$ of each GCN layer $h^l$ are set to 128, The dimension of the final feature $H$ of the Structure View is set to 768. For loss function, the hyperparameter $\alpha$ is set to 0.01, $\beta$ is set to 0.1.

**Training and Evaluation.** Adam optimization [47] is adopted to optimize our model. We use Adam optimization with a default learning rate of $2 \times 10^{-4}$, weight decay of $1 \times 10^{-5}$, and the batch size is set to 8. All experiment results are obtained through 5-fold cross-validation. Concordance index (C-index) [48] and its standard deviation (std) are used to measure the predictive performance in correctly ranking the survival risk of each patient. As qualitative assessment, we use Kaplan-Meier curves [49] to visualize the quality of patient stratification in stratifying low and high-risk patients as two different survival distributions. All the experiments are implemented using PyTorch [50] on a workstation with 4 Nvidia 3090 GPUs.

## 5.3 Comparison with State-Of-The-Art Methods

We compare our proposed method with several state-of-the-art survival prediction approaches on the above datasets. The comparison methods include: (1) WSISA [19], (2) ABMIL [4], (3) TransMIL [51], (4) DeepAttnMISL [52], (5) Patch-GCN [5], (6) DeepGraphConv [53], (7) HEAT [23], (8) HGSurvNet [54]. (9) PANTHER [55]. Among them, WSISA, ABMIL, TransMIL and DeepAttnMISL are classic WSI survival prediction methods; DeepGraphConv and Patch-GCN used GNN to construct homogeneous graphs; HGSurvNet established a hypergraph; HEAT created a heterogeneous graph for classification/staging tasks; PANTHER utilizes an unsupervised prototype network to aggregate global information. For fairness of comparison, we use UNI [35] as the instance feature encoder for all methods. All approaches are evaluated using the same 5-fold cross-validation splits.

**Comparison.** From the results in table 1, we observe that ProtoSurv achieves best or suboptimal performance across five cancer datasets. Particularly, the PAAD dataset has a small sample size (208 cases), which may lead to overfitting. Therefore, PANTHER, which extracts prototypes in an unsupervised way based on priors and has a minimal number of learnable parameters, demonstrated

---

[1]https://portal.gdc.cancer.gov/

superior results on PAAD dataset. Meanwhile, ProtoSurv is only slightly below PANTHER on PAAD and significantly outperforms the other comparative methods, indicating the potential of the domain-prior infused model on small-scale datasets.

Table 1: C-index (mean ± std) over five cancer datasets. The best and second-best results are highlighted in **bold** and underlined.

|  | PAAD | BRCA | LGG | LUAD | COAD |
|---|---|---|---|---|---|
| WSISA | 0.573 ± 0.021 | 0.564 ± 0.054 | 0.610 ± 0.013 | 0.576 ± 0.045 | 0.564 ± 0.034 |
| ABMIL | 0.625 ± 0.063 | 0.657 ± 0.064 | 0.710 ± 0.048 | 0.653 ± 0.059 | 0.647 ± 0.036 |
| TransMIL | 0.642 ± 0.037 | 0.694 ± 0.053 | 0.739 ± 0.034 | 0.608 ± 0.040 | **0.695 ± 0.051** |
| DeepAttMISL | 0.596 ± 0.034 | 0.634 ± 0.017 | 0.657 ± 0.076 | 0.623 ± 0.049 | 0.638 ± 0.069 |
| Patch-GCN | 0.618 ± 0.057 | 0.647 ± 0.032 | 0.713 ± 0.054 | 0.635 ± 0.027 | 0.652 ± 0.086 |
| DeepGraphConv | 0.615 ± 0.032 | 0.535 ± 0.014 | 0.617 ± 0.048 | 0.597 ± 0.037 | 0.621 ± 0.085 |
| HEAT | 0.638 ± 0.030 | 0.693 ± 0.084 | 0.741 ± 0.079 | 0.642 ± 0.031 | 0.679 ± 0.056 |
| HGSurvNet | 0.646 ± 0.064 | 0.701 ± 0.067 | 0.746 ± 0.043 | 0.638 ± 0.064 | 0.673 ± 0.043 |
| PANTHER | **0.673 ± 0.082** | 0.699 ± 0.019 | 0.748 ± 0.046 | 0.631 ± 0.029 | 0.635 ± 0.056 |
| ProtoSurv (Ours) | 0.669 ± 0.049 | **0.720 ± 0.040** | **0.774 ± 0.063** | **0.658 ± 0.046** | 0.692 ± 0.045 |

## 5.4 Interpretability

To understand the interaction patterns of the prototypes, we visualize the attention heatmap between the prototypes and the features, as well as the attention components for each prototype in fig. 2. We observed that the attention preferences of multi-prototypes from a category varied. Some prototypes were responsible for extracting global category information, while others focused on discovering interactions between other categories. This indicates that the prototype learning paradigm has the potential to uncover unknown interactions and factors.

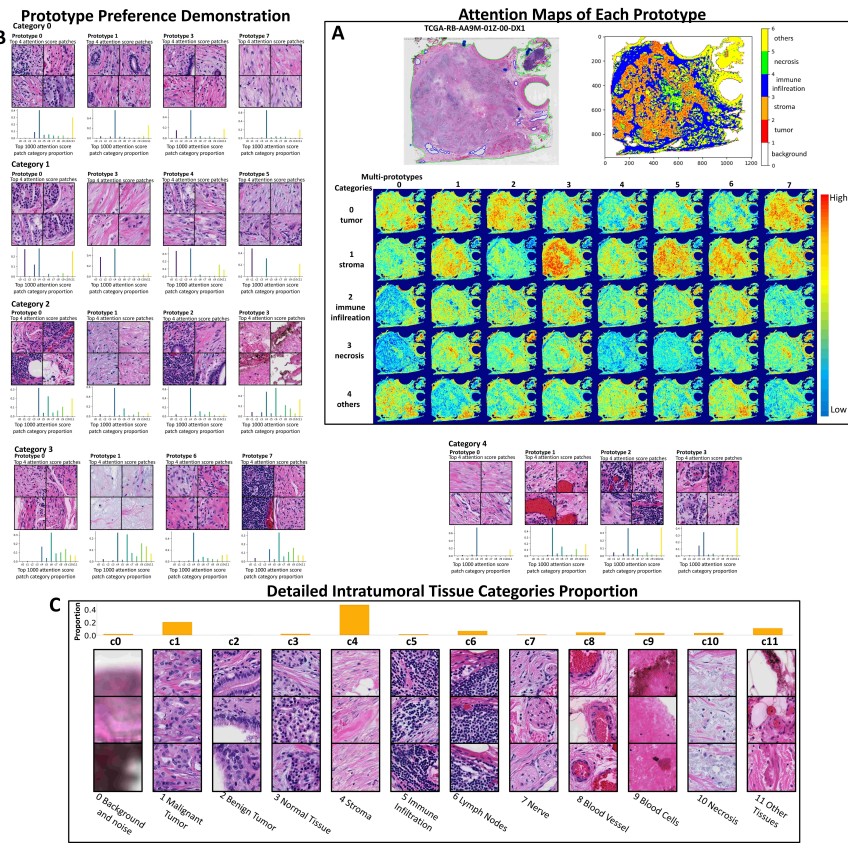

Figure 2: **Heatmap interpretation of prototypes.** (A) Visualizations of attention maps of multi-prototypes in Prior Guided Fusion (PGF) module. (B) Visualization of each prototype's preference: showing the top 4 patches and detailed tissue proportion of the top 1000 patches base on attention scores. (C) Proportions of detailed 12 intratumoral tissue categories.

## 5.5 Ablation Studies

**Component Validation.** We decouple the graph into the Structure View and the Histology View to separately extract structure and global features, and use the Prior Guided Fusion & Pooling module to aggregate these features. In this section, we conduct ablation experiments on Histology View (HV), Structure View (SV), and Prior Guided Fusion & Pooling (PGF) modules. Specifically, we implement our ablation as follows: (1) Without Histology View (w/o HV): We retain the features after message passing in the Structure View in eq. (3). Same to PatchGCN [5], we compute attention scores for each node and aggregate them by weighted summation. (In fact, the ablated model without HV is equivalent to PatchGCN.) (2) Without Structure View (w/o SV): We use the multi-prototypes extracted from the Histology View in eq. (7). We test the use of mean pooling and direct concatenation (concat pooling) to obtain the WSI-level representation. (3) Without Prior Guided Fusion & Pooling (w/o PGF): Beside our proposed fusion component, we tested two other aggregation approaches, including direct concatenation fusion and the transpose fusion. Within the concatenation fusion, we directly concatenate the prototypes from HV with the features from SV. Within the transpose fusion, We use the aggregation method from Dong et al. [16], swapping the order of $Q$ and $KV$ in cross-attention to integrate global information into node features, and same to the ablation of HV module, we compute attention scores for each node feature and aggregate them by weighted summation. For detailed information on the ablation of the PGF module, please refer to appendix D.1. Table 2 presents the results. We find that combining the HV and SV modules resulted in improvements, regardless of the aggregation method. This indicates that HV and SV are compatible and complementary. Additionally, our proposed Prior Guided Fusion (PGF) can best utilize information from the two modules to achieve optimal aggregation.

Table 2: Ablation study of the main modules in ProtoSurv.

|  | PAAD | BRCA | LGG | LUAD | COAD |
|---|---|---|---|---|---|
| w/o HV | $0.618 \pm 0.057$ | $0.647 \pm 0.032$ | $0.713 \pm 0.054$ | $0.635 \pm 0.027$ | $0.652 \pm 0.086$ |
| w/o SV(mean pooling) | $0.624 \pm 0.032$ | $0.657 \pm 0.049$ | $0.706 \pm 0.036$ | $0.646 \pm 0.051$ | $0.684 \pm 0.044$ |
| w/o SV(concat pooling) | $0.661 \pm 0.057$ | $0.713 \pm 0.039$ | $0.766 \pm 0.044$ | $0.641 \pm 0.037$ | $0.688 \pm 0.046$ |
| w/o PGF(transpose fusion) | $0.653 \pm 0.024$ | $0.714 \pm 0.041$ | $0.724 \pm 0.042$ | $0.653 \pm 0.055$ | $0.657 \pm 0.074$ |
| w/o PGF(concat fusion) | $0.652 \pm 0.040$ | $0.719 \pm 0.024$ | $0.712 \pm 0.084$ | $0.662 \pm 0.064$ | $0.659 \pm 0.058$ |
| ProtoSurv | $0.669 \pm 0.049$ | $0.720 \pm 0.040$ | $0.774 \pm 0.063$ | $0.658 \pm 0.046$ | $0.692 \pm 0.045$ |

**Classifiers.** Classifying each patch into tissue categories requires additional annotated data to train the classifier, which somewhat limits the applicability of our model. To demonstrate the performance of our model in a wider range of scenarios, we test the performance of the model with different publicly available tissue classifiers in this section. We test our model using four different classification methods: (1) Zero-shot classifier from CONCH [56] (CONCH), (2) HoverNet nuclear classification [57] (HoverNet), (3) Pre-compute initial prototypes $P_{init}^c$ of eq. (4) from the features of existing patches of category $c$ (Pre-Proto), (4) K-means (with cluster number n = 4, 6, 8). None of the classifiers are specifically optimized for our task. For details on the usage of these classifiers, refer to appendix D.2. Table 3 shows the results of ProtoSurv under different patch classifiers. We find that without specially optimized classifiers, ProtoSurv still exhibited excellent performance. To our surprise, even when simply replacing the classifier with K-means clustering, ProtoSurv still outperforms most comparative methods. The ablation experiments on classifiers demonstrate the robustness of ProtoSurv to classifier choice, showing that ProtoSurv can achieve state-of-the-art performance without the need for specialized classifiers.

Table 3: Effect of different classifiers.

|  | PAAD | BRCA | LGG | LUAD | COAD |
|---|---|---|---|---|---|
| CONCH | $0.664 \pm 0.051$ | $0.729 \pm 0.042$ | $0.776 \pm 0.051$ | $0.660 \pm 0.056$ | $0.692 \pm 0.040$ |
| HoverNet | $0.649 \pm 0.060$ | $0.701 \pm 0.070$ | $0.771 \pm 0.064$ | $0.656 \pm 0.042$ | $0.695 \pm 0.036$ |
| Pre-Proto | $0.668 \pm 0.045$ | $0.714 \pm 0.062$ | $0.775 \pm 0.062$ | $0.652 \pm 0.055$ | $0.687 \pm 0.038$ |
| K-means(n=4) | $0.646 \pm 0.056$ | $0.683 \pm 0.080$ | $0.774 \pm 0.069$ | $0.643 \pm 0.051$ | $0.690 \pm 0.031$ |
| K-means(n=6) | $0.652 \pm 0.027$ | $0.708 \pm 0.049$ | $0.772 \pm 0.061$ | $0.638 \pm 0.046$ | $0.695 \pm 0.037$ |
| K-means(n=8) | $0.656 \pm 0.039$ | $0.696 \pm 0.054$ | $0.761 \pm 0.070$ | $0.648 \pm 0.057$ | $0.664 \pm 0.045$ |

**Choice of Tissue Categories.** To infuse our model with prior knowledge, we select node categories based on prior prognosis-related tissues. These ablation experiments are conducted on different tissue categories to validate the effectiveness of incorporating prior knowledge. We train the classifier to categorize 12 types of tissues, which are: 0. background and noise; 1. malignant tumor; 2. benign tumor; 3. normal tissue; 4. stroma; 5. immune infiltration; 6. lymph nodes; 7. nerve; 8. blood vessel; 9. blood cell aggregation; 10. necrosis; 11. other tissues. We refer to these 12 detailed tissue categories as "Detailed Tissue Category (DTC)". Additionally, based on histological knowledge, we coarsely group these 12 tissue categories into 5 broader categories (The numbers in parentheses correspond to the numbers of DTC): Non-tumor tissue (3); tumor (1,2); stroma(4,5,7,8); necrosis(10); others(0,6,9). We refer to these 5 broader categories as "Coarse Tissue Category (CTC)". In our model, we incorporate five tissue categories based on pathological knowledge of prognosis-related tissues: tumor(1,2); stroma(4); immune infiltration(5); necrosis(10); others(0,3,6,7,8,9,11). We refer to these 5 categories from prior knowledge as "Prior Tissue Category (PTC)". Table 4 shows the results of ProtoSurv under three different tissue category settings. We observed that the detailed category setting generally has a negative impact on the model's performance. We believe this is due to the overly detailed categories introducing a substantial amount of irrelevant information to the model. Compared to the Coarse Tissue Category, our proposed Prior Tissue Category achieved the best results, demonstrating the effectiveness of incorporating prior knowledge.

Table 4: Effect of tissue category choices.

|  | PAAD | BRCA | LGG | LUAD | COAD |
|---|---|---|---|---|---|
| DTC | $0.641 \pm 0.087$ | $0.684 \pm 0.063$ | $0.793 \pm 0.056$ | $0.652 \pm 0.064$ | $0.681 \pm 0.051$ |
| CTC | $0.656 \pm 0.066$ | $0.706 \pm 0.058$ | $0.776 \pm 0.064$ | $0.658 \pm 0.050$ | $0.690 \pm 0.041$ |
| PTC(proposed) | $0.669 \pm 0.049$ | $0.720 \pm 0.040$ | $0.774 \pm 0.063$ | $0.658 \pm 0.046$ | $0.692 \pm 0.045$ |

**Further Ablations**. We conduct further ablation studies and present additional insights in **appendix C**. Overall, ProtoSurv is robust of errors in classification, hyperparameters of the losses and the number of prototypes. Additionally, we provide statistics on FLOPs and model runtime, and test the complexity and performance of the tiny version of the model. The results indicate that although there is an increase in runtime compared to classical models, it remains acceptable, and the model still achieves decent performance with smaller parameter settings.

## 6    Discussions

**Conclusion.** In this paper, we introduce a heterogeneous graph for WSI survival prediction to incorporate pathological prior knowledge by leveraging tissue heterogeneity in tumor, and propose a novel heterogeneous graph message passing framework (ProtoSurv) to integrate pathological prior knowledge into the model. Compared to previous work, ProtoSurv is not solely data-driven but learns under the guidance of prior expert knowledge. We validate our method across multiple cancer datasets, demonstrating that ProtoSurv exhibits higher accuracy and robustness, as well as more stable multi-cancer performance.

**Limitations and Future Work.** Although ablation experiments have demonstrated the robustness of ProtoSurv across different classifiers, the difficulty of obtaining node categories remains an obstacle to its broader application. In future work, we will explore methods to obtain pseudo-labels directly from node features without relying on additional classifiers.

## Acknowledgments and Disclosure of Funding

This study is supported by the National Natural Science Foundation of China (82302316). The authors declare that they have no known competing financial interests or personal relationships that could have appeared to influence the work reported in this paper.

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

# A  Overview

In this supplement, We first provide detailed illustrations of ProtoSurv in appendix B. We then provide further ablation studies in appendix C. Then, we provide implementation details of ablation studies in appendix D. We display the KM curve results of ProtoSurv on five datasets in appendix E, and present different subtypes within the intratumoral tissue in pathology in appendix F to support the motivation for our model's multi-prototype design. Finally, we discuss the potential ethical issues that may arise in our study in appendix G.

# B  Detailed Illustrations of ProtoSurv

In this section, we provide schematic diagrams illustrating the details of each module's functionality. In fig. 3, we illustrate the schematic diagram of the GCN providing multi-hop neighborhood information. In fig. 4, we illustrate the process of obtaining multi-prototypes.

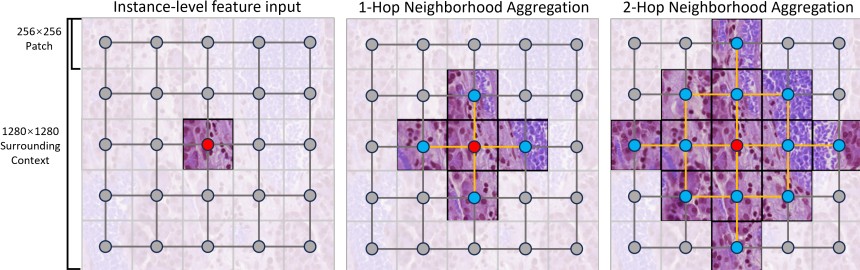

Figure 3: Illustration of the multi-hop neighborhood information within the Structure View.

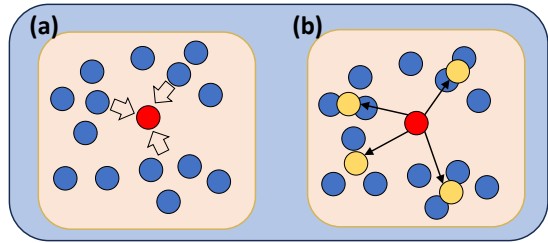

Figure 4: Illustration of the multi-prototypes within the Histology View. (a) Calculate the initial prototype $p_{init}^c$ of a certain category using the average function. (b) Using learnable parameters to shift $p_{init}^c$ to multi-prototypes $P_{prior}^c = \{p_c^1, p_c^2, \ldots, p_c^k\}$, to focus on different clusters (subtypes) within the category.

# C  Further Ablations

**Extract Features only using the last or last two Layers in Structure View.** We tested using only the last and the last two layers to extract features in SV. As shown in Table 5, although the optimal results varied across different datasets, overall, models that used more layers to extract features achieved better results.

Table 5: Effects of GCN layer number in Structure View.

|  | PAAD | BRCA | LGG | LUAD | COAD |
|---|---|---|---|---|---|
| ProtoSurv (SV all layers) | $0.669 \pm 0.049$ | $0.720 \pm 0.040$ | $0.774 \pm 0.063$ | $0.658 \pm 0.046$ | $0.692 \pm 0.045$ |
| ProtoSurv (SV last layer) | $0.671 \pm 0.042$ | $0.718 \pm 0.049$ | $0.764 \pm 0.054$ | $0.658 \pm 0.060$ | $0.678 \pm 0.051$ |
| ProtoSurv (SV last two layers) | $0.662 \pm 0.049$ | $0.723 \pm 0.044$ | $0.762 \pm 0.037$ | $0.656 \pm 0.058$ | $0.693 \pm 0.057$ |

**Number of Prototypes per Category.** To guide the model to focus on different subtype features of a specific tissue category, we assign multiple prototypes for each category. Our model hypothesizes

that using multiple prototypes can capture the feature distribution of different subtypes within certain tissue category. We vary the number of prototypes per class to validate the contribution of multiple prototypes to the results. Fig.5 illustrates ablative experiments of ProtoSurv on PAAD dataset. We observe that: (1) ProtoSurv with a single prototype per class performs almost on par with state-of-the-art methods. (2) ProtoSurv with multiple prototypes per class consistently outperforms the version with a single prototype. The ablative experiments show the effectiveness of using prototypes, as well as the benefits of using multiple prototypes per class over using a single one.

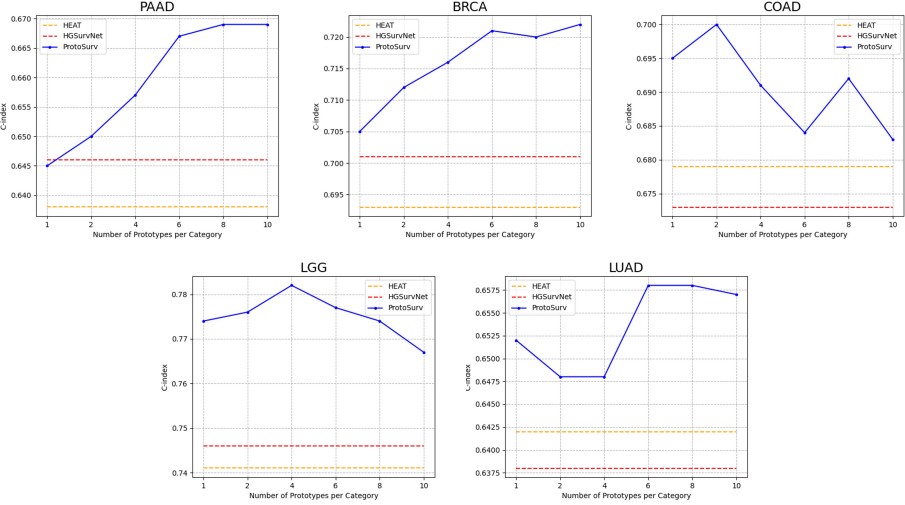

Figure 5: Effect of prototype number per category.

**How Errors in Classification Impact the Overall Performance.** To minimize the impact of classification errors on the overall performance, in the HV module, we rely only on the node categories provided by the classifier to delineate an initial range. We calculate the average value of the patch features within the category range as the initial prototype, providing an initial preference for the aggregation of each prototype (pathology prior injection). To further illustrate this point, we randomly assign categories for 20% and 30% of the nodes. As shown in table 6, the introduction of random noise does not significantly impact the model's performance, indicating the robustness of our model against classification errors.

Table 6: Effects of classification errors.

|  | PAAD | BRCA | LGG | LUAD | COAD |
|---|---|---|---|---|---|
| ProtoSurv | $0.669 \pm 0.049$ | $0.720 \pm 0.040$ | $0.774 \pm 0.063$ | $0.658 \pm 0.046$ | $0.692 \pm 0.045$ |
| 20% random | $0.671 \pm 0.044$ | $0.712 \pm 0.045$ | $0.769 \pm 0.048$ | $0.658 \pm 0.046$ | $0.685 \pm 0.051$ |
| 30% random | $0.666 \pm 0.047$ | $0.717 \pm 0.047$ | $0.777 \pm 0.047$ | $0.656 \pm 0.043$ | $0.689 \pm 0.053$ |

**Loss Functions.** To fully exploit the capabilities of the multiple prototypes of each tissue categories, we introduce compatibility loss and orthogonality loss to regularize training. In this ablation study, we evaluate the effect of these losses on model performance. Table 7 shows the results of ProtoSurv under different settings of hyperparameters $\alpha$ and $\beta$. Compared to cox loss alone, performance improvements for all non-zero values of $\alpha$ and $\beta$ on both datasets. This provides substantial evidence for the usefulness of the additional loss functions.

Table 7: Effects of compatibility and orthogonality losses.

| $\mathcal{L}_{comp}(\alpha)$ | $\mathcal{L}_{ortho}(\beta)$ | PAAD | BRCA | LGG | LUAD | COAD |
|---|---|---|---|---|---|---|
| 0 | 0 | $0.651 \pm 0.057$ | $0.693 \pm 0.046$ | $0.765 \pm 0.054$ | $0.661 \pm 0.052$ | $0.690 \pm 0.046$ |
| 0.1 | 0.1 | $0.656 \pm 0.049$ | $0.702 \pm 0.052$ | $0.769 \pm 0.051$ | $0.654 \pm 0.046$ | $0.694 \pm 0.038$ |
| 0.1 | 0.01 | $0.658 \pm 0.055$ | $0.723 \pm 0.037$ | $0.773 \pm 0.063$ | $0.654 \pm 0.041$ | $0.692 \pm 0.043$ |
| 0.01 | 0.1 | $0.669 \pm 0.049$ | $0.720 \pm 0.040$ | $0.774 \pm 0.063$ | $0.658 \pm 0.046$ | $0.692 \pm 0.045$ |
| 0.01 | 0.01 | $0.662 \pm 0.043$ | $0.714 \pm 0.025$ | $0.770 \pm 0.060$ | $0.658 \pm 0.049$ | $0.690 \pm 0.044$ |

**Computational Requirements.** We evaluate the model's inference time, floating point of operations(FLOPs), model parameters, and maximum GPU memory usage. We use a WSI which contains 32,625 patches as input. The computation time is measured using a Nvidia RTX 3090 GPU. We included PatchGCN [5] for comparison. We additionally test ProtoSurv-tiny under a reduced parameter configuration (prototype dim = 256, hidden dim of SV and HV = 64, prototypes per category = 4), to evaluate the performance degradation of our architecture with fewer parameters and its scalability for more limited hardware. Table 8 presents the computational requirements of the model, while table 9 shows the performance of the model under the tiny setting.

Table 8: Computational requirements.

|  | Time (s) | FLOPs (G) | Model Parameters (M) | Maximum GPU memory usage (MB) |
|---|---|---|---|---|
| ProtoSurv | 0.29 | 627.3 | 39.1 | 5417 |
| ProtoSurv-tiny | 0.21 | 96.5 | 4.77 | 1523 |
| PatchGCN | 0.12 | 30.5 | 1.19 | 1570 |

Table 9: Performance of ProtoSurv-tiny.

|  | PAAD | BRCA | LGG | LUAD | COAD |
|---|---|---|---|---|---|
| ProtoSurv | $0.669 \pm 0.049$ | $0.720 \pm 0.040$ | $0.774 \pm 0.063$ | $0.658 \pm 0.046$ | $0.692 \pm 0.045$ |
| ProtoSurv-tiny | $0.687 \pm 0.049$ | $0.707 \pm 0.044$ | $0.756 \pm 0.038$ | $0.664 \pm 0.039$ | $0.673 \pm 0.039$ |

# D   Implementation Details of Ablation Studies

## D.1   Implementation Details of Component Ablation

In this section, we provide a detailed explanation of the ablation settings for the Prior Guided Fusion & Pooling (PGF) module in the component validation ablation experiment. The ablation of the PGF module aggregates the global prototypes into the graph nodes, here the eq. (8) is modified to:

$$H = \text{Softmax} \left( \frac{\left[ HW^Q \right] \left[ PW^K \right]^{\text{T}}}{\sqrt{d}} \right) \left( PW^V \right) \in \mathbb{R}^{N \times d}, \tag{15}$$

Then, we use the weighted aggregation method from Chen et al. [5] to calculate the score for each node and combine them with weighted summation.

$$a_i = \frac{\exp \left\{ w^T \left( tanh \left( MH_i \right) \odot \text{sigm} \left( UH_i^T \right) \right) \right\}}{\sum\limits_{i=1}^{K} \exp \left\{ w^T (tanh(MH_i) \odot \text{sigm}(UH_i^T)) \right\}}, \tag{16}$$

$$h_{slide} = \sum_{i=1}^{K} a_i H_i, \tag{17}$$

$$Y = MLP(h_{slide}). \tag{18}$$

This weighted aggregation method is also used in the ablation of the Histology View (HV) module.

## D.2   Implementation Details of Classifier Ablation

**CONCH.** CONCH [56] is a visual-language foundation model. The model can be immediately used for downstream classification tasks due to the aligned visual-language pretraining, which eliminates the need for additional labeled examples for supervised learning or fine-tuning. As shown in fig. 6, we use text prompts to map patches and prompts into the same embedding space, comparing the cosine similarity of the representations to obtain category labels. We constructed our set of predetermined text prompts based on the set of class or category names provided by CONCH. The set of categories and their names we use including: 0. lymphoid infiltrate 1. stroma 2. tumor 3. necrosis 4. others (adipose, background, penmarking, mucin, muscle, benign epithelium)

**HoverNet.** Follow Chan et al. [23], we test using HoverNet [57] as the classifier in the ablation experiment. HoverNet detects nuclei in each patch and assigns types to these nuclei. By majority

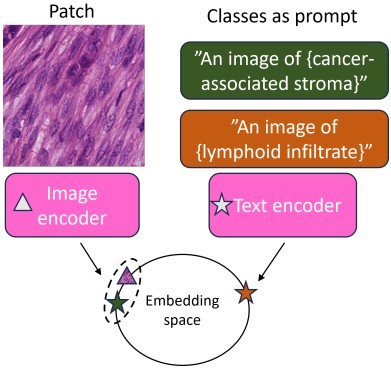

Figure 6: Schematic of zero-shot classification using contrastively aligned image and text encoders of CONCH.

votes, we take the most frequently predicted nucleus type to be the type of the patches. The nuclei types of HoverNet including: neoplastic, dead, inflammatory, non-neoplastic epithelial, connective, no label.

**Pre-Proto.** In the ablation study of pre-proto, we pre-computed the initial prototypes $P_{init}^c$ for each category $c$ based on the representations of existing category patches. This eliminates the need for the model to get the category of each node for every sample. However, it reduces the model's attention to the specificity of the tissues in each sample. In this ablation experiment, for each cancer type, we randomly sampled 50% of the WSIs and then randomly sampled 10% of the patches within those WSIs to calculate their representations and corresponding tissue categories. Then, we used the average function to calculate the initial prototype $P_{init}^c$ for each category of patches. Under this setting, all samples use the same pre-obtained prototypes, and the Histology View starts directly from eq. (5).

## E   Kaplan-Meier Curves

We evaluate our method using the Kaplan-Meier curves as presented in fig. 7. In Kaplan-Meier analysis, patients are separated into high-risk and low-risk groups based on predicted risk scores. We use the median value of each validation set as the cut-off. Subsequently, we utilize the log-rank test to compute P-values, which assess the statistical significance of differences between these groups. The results indicate that our method's predictions are statistically significant.

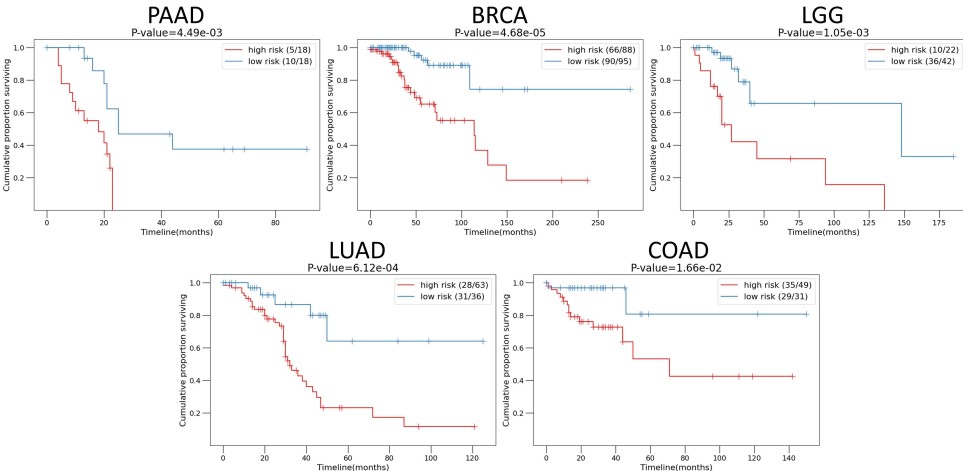

Figure 7: Kaplan-Meier curves of predicted high-risk (red) and low-risk (blue) groups. A P-value <0.05 indicates statistical significance.

# F Illustrations of subtypes of tissue categories

In this section, we present illustrations of various subtypes of certain tissues to support our use of multi-prototypes. fig. 8 illustrate subtypes of stroma; fig. 9 illustrate subtypes of immune infiltration; fig. 10 illustrate subtypes of tumor.

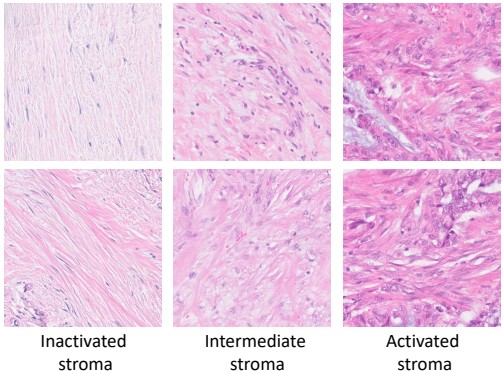

Figure 8: Subtypes of stroma [58].

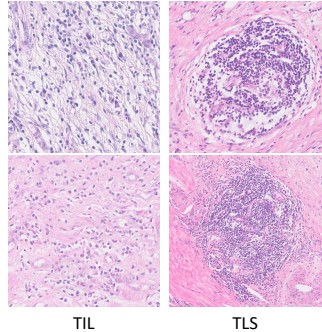

Figure 9: Subtypes of immune infiltration.

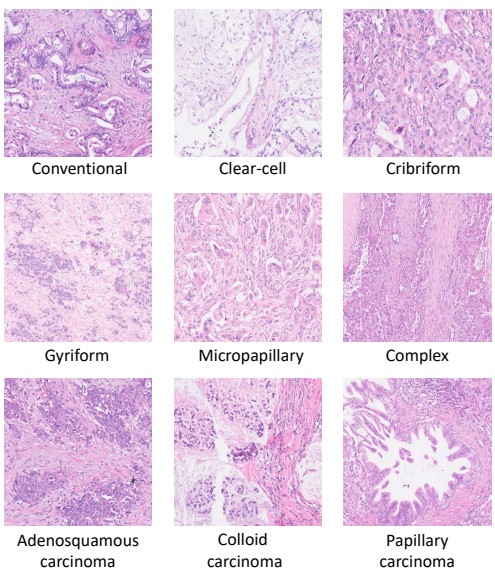

Figure 10: Subtypes of tumor [59].

# G   Ethical Discussions

**Ethical Issues.** With regard to possible ethical issues in data collection, The Cancer Genome Atlas (TCGA) repository, as a publicly available dataset that has been used in pathology previous studies, is undoubtedly not ethically questionable. Additionally, we annotated tissue categories for a subset of slices from the TCGA dataset to train the patch classifier. The pathologists involved in the annotation were explicitly informed of the purpose of sample collection, ensuring it would not adversely affect any individuals. Therefore, it does not adversely affect any individual, so there are no ethical or moral issues.

**Possible Negative Social Impacts.** As the research in this paper deals with the survival prediction of cancer, it is necessary to elaborate here on the possible negative social impacts of this work. Including but not limited to:

- Incorrect diagnosis. AI methods must have the possibility of error, which cannot be avoided, but an incorrect diagnosis will have a significant impact on individuals and society. Therefore, AI tools can only be used as a diagnostic aid, not as a decision maker, and the final decision should still be made by the doctor.

- Leakage of privacy information. In WSI datasets, the identity information of the subjects is highly private, and the leakage of identity information will also have unpredictable and significant impact on individuals and society. Therefore, in this work, we exclusively used WSI data from public datasets, where their privacy information has been well protected.

