# OpenReview forum: "Leveraging Tumor Heterogeneity: Heterogeneous Graph Representation Learning for Cancer Survival Prediction in Whole Slide Images"
_NeurIPS.cc/2024/Conference — NeurIPS 2024 poster_

### Official Review · Reviewer_S12V · 2024-07-10

**Soundness:** 3
**Presentation:** 3
**Contribution:** 3
**Rating:** 5
**Confidence:** 4

**Summary:**

The authors present ProtoSurv, a model that leverages heterogneous graph representation learning to predict survival risk. Their model can be decoupled into two basic submodules: 1)Structrure view module whish is responsible for injectiing the topological ingormation of the wsi into the model and the 2) histology view module  which incorporates pathological priors reflecting tumor heterogeneity. The authors conducted experiments across five WSI datasets, demonstrating superior performance, supported a series of by ablation studies.

**Strengths:**

- The paper is clear and well-written.
- The methodology that is introduced is novel, tackling the inherent heterogeneity of pathology images
- There is a detailed series of abalation studies justifying the use of each component in the model

**Weaknesses:**

- The prototyes that are used in histology view are not learnt, but are rather selected before training based on prior knowledge, potentially hindering the adaptability of the model since preselected prototypes may not capture the full variability or complexity of the data. This could also introduce generalisation issues because the model may struggle to generalize to new, unseen data that differ significantly from the prototypes.

**Questions:**

- Andrew H. Song et al. have recentrly published PANTHER [Morphological Prototyping for Unsupervised Slide Representation Learning in Computational Pathology](https://openaccess.thecvf.com/content/CVPR2024/papers/Song_Morphological_Prototyping_for_Unsupervised_Slide_Representation_Learning_in_Computational_Pathology_CVPR_2024_paper.pdf)  that explores a similar idea of extracting morphological signatures using protypes from WSIs in an unsupervised way showcasing superior results. Have you considered comparing PANTHER to your model?

**Limitations:**

- Apart from the comparisons with SOTA methods, it would be interesting to shift the focus also to the interepretability of the model. The prototypes are instrumental in enhancing the model's performance, offering insight into how prototype-guided decisions contribute to the predicted outcome. There is only one attention map in the suplementary material.

---

> ### Author Rebuttal · Authors · 2024-08-07
>
> Thanks for your constructive comment. Here are the responses to Weaknesses (W), Questions (Q) and Limitations(L).
>
> **W. The prototyes that are used in histology view are not learnt,
> potentially hindering the adaptability of the model since preselected prototypes may not capture the full variability or complexity of the data.**
>
> We fully agree with your comment.
> We also consider that prototypes selected before training based on prior knowledge indeed may struggle to capture the full variability or complexity of the data.
> Therefore, when designing the prototype extraction module (the Histology View (HV) module),
> we rely solely on the node categories provided by prior knowledge to give each prototype an initial preference.
> Then we use learnable shifts to obtain diverse prototypes for each category,
> enabling them to prefer on different factors within the category,
> including both previously known and unknown factors / phenotypes.
> Finally, under the guidance of these preferences,
> the prototypes to aggregate relevant information from the global features by learnable cross-attention.
> In summary, our prototype extraction process is learnable and highly flexible,
> and has the ability to capture various previously unknown factors / phenotypes that may contribute to patient outcomes.
>
> We apologize for any misunderstanding caused by the description in our paper.
> We will revise this section in the camera-ready version to more clearly explain our prototype extraction process.
> Additionally, we have uploaded a detailed interpretability figure in the PDF attachment in "Author Rebuttal",
> which visualizes the attention preferences of multiple prototypes across different categories to further support our claims.
> If you are interested about it, please refer to "Author Rebuttal" for more details.
>
> **Q. Compare with PANTHER.**
>
> Thanks for the advice.
> We compared PANTHER with our model.
> Additionally, we compared our model's Histology View (HV) module (prototype extraction module) to ensure the fairness of the prototype extraction module comparison.
> We tested two pooling methods for the HV module,
> including the mean pooling method used in the ablation experiments in the main paper (corresponding to the "without (w/o) HV" row in Table 2 in the main paper),
> as well as the concat pooling method used by PANTHER (PANTHER performs pooling by concatenating all prototypes along the channel dimension).
> The experimental results are as follows:
>
> |                    |       COAD        |        LGG        |       LUAD        |       PAAD        |       BRCA        |
> |:------------------:|:-----------------:|:-----------------:|:-----------------:|:-----------------:|:-----------------:|
> |     ProtoSurv      | **0.692 ± 0.045** | **0.774 ± 0.063** | **0.658 ± 0.046** |   0.669 ± 0.049   | **0.720 ± 0.040** |
> |      PANTHER       |   0.635 ± 0.056   |   0.748 ± 0.046   |   0.631 ± 0.029   | **0.673 ± 0.082** |   0.699 ± 0.019   |
> |  HV(mean pooling)  |   0.684 ± 0.044   |   0.706 ± 0.036   |   0.646 ± 0.051   |   0.624 ± 0.032   |   0.657 ± 0.049   |
> | HV(concat pooling) |   0.688 ± 0.046   |   0.766 ± 0.044   |   0.641 ± 0.037   |   0.661 ± 0.057   |   0.713 ± 0.039   |
>
> The learnable prototype extraction method (HV module) outperforms PANTHER on most datasets.
> In addition, we observed that PANTHER's performance surpassed all comparison models on the PAAD dataset.
> We attribute this to the relatively small sample size of the PAAD dataset (208 cases), which may lead to overfitting.
> Therefore, PANTHER, which extracts prototypes in an unsupervised way based on priors and has a minimal number of learnable parameters,
> demonstrated superior results on the PAAD dataset.
>
> **L. Apart from the comparisons with SOTA methods, it would be interesting to shift the focus also to the interpretability of the model.**
>
> Thank you for your interest in the interpretability of our model.
> We have expanded the figure to visualize the preferences of each prototype and the tissues they focus on,
> to better demonstrate the interpretability of our model.
> if you are interested about it, please refer to "Author Rebuttal" for more details.
>
> Once again, thank you for your careful review and valuable comments on our paper.
> If you are satisfied with our response,
> would you kindly bump up your score?

---

### Official Review · Reviewer_gRNp · 2024-07-12

**Soundness:** 2
**Presentation:** 3
**Contribution:** 2
**Rating:** 5
**Confidence:** 5

**Summary:**

This paper analyzes the limitations of the existing MIL method in survival prediction with WSIs 1) overfitting, 2) numerous redundant and irrelevant instances, and 3) insufficient exploration of the interaction between local, regional features and global contextual features in WSI. To address these issues, the paper proposes Multiple Instance Learning with Hierarchical Graph Transformer over Super-Pixel (HGTSP). HGTSP consists of three novel modules 1) a Pixel-based Pseudo-Bag Division (SPBD), 2) a Super-Pixel Region Sampling (SPRS), and 3) a Hierarchical Graph Transformer (HGT). The experiments on TCGA verify the effectiveness of HGTSP. However, the modules proposed in this paper only partially correspond to the issues it claims to solve. In addition, HGTSP contains a large number of hyper-parameters, which makes it challenging to apply this method to new datasets quickly.

**Strengths:**

The limitations of previous methods have been well analyzed: 1) overfitting, 2) numerous redundant and irrelevant instances, and 3) insufficient exploration of the interaction between local, regional features and global contextual features in WSI. To address these limitations, this paper proposes 1) a Pixel-based Pseudo-Bag Division (SPBD), 2) a Super-Pixel Region Sampling (SPRS), and 3) a Hierarchical Graph Transformer (HGT). The experiments on TCGA verify the effectiveness of HGTSP.

**Weaknesses:**

1.	This paper claims to address the overfitting via SPBD in the abstract. However, according to the introduction, SPBD is mainly used to address the risk of pseudo-bag mislabeling and inconsistency between original bags and pseudo-bags. Therefore, it would be better to refine the claims in the abstract.
2.	There are a large number of hyper-parameters. According to the appendix, these hyper-parameters highly influence the performance of HGTSP.

**Questions:**

1.	As HGTSP combines multiple techniques, including K-means clustering, adaptive sampling, and Graph Transformer, comparing the time required for inferring the same number of samples is more appropriate than only comparing FLOPs and the number of parameters.
2.	Is there a method to automatically set hyper-parameters?

**Limitations:**

Yes

---

> ### Comment · Reviewer_gRNp · 2024-08-01
> **update review**
>
> I'm sorry for the reviews submitted for another paper. I update the reviews here.
>
>
>
> This paper proposes ProtoSurv, a heterogeneous graph model for WSI survival prediction. ProtoSurv is driven by data and incorporates pathological domain knowledge. Specifically, ProtoSurv consists of three modules 1) a multi-layer GCN to learn the structure representation of WSIs, 2) a prototype representation method to learn pathological priors, 3) a prior guided fusion method to aggregate structure view features and pathological multi-prototypes. The experiments on TCGA verify the effectiveness of ProtoSurv. The motivation of this paper sounds good. However, the model lacks novelty and in-depth exploration. In addition, ProtoSurv requires additional labels to train a classifier, leading to unfair comparison.
>
> Soundness: 3: good
> Presentation: 3: good
> Contribution: 2: fair
> Strengths:
> This paper is easy to follow.
> The motivation sounds good.
> This paper proposes ProtoSurv, which deciphers intratumoral tissue heterogeneity using a heterogeneous graph and incorporates prior knowledge of prognostic tissue types into the prediction process.
> The experiments on TCGA verify the effectiveness of ProtoSurv.
> Weaknesses:
> The design of the main modules lacks novelty and in-depth exploration. The Structure View (SV) is a type of multi-layer feature aggregation that has been widely studied and may be highly influenced by the selection of layers to extract features. The Histology View (HV) is a type of feature shifting which has also been widely studied.
> ProtoSurv requires additional labels to train a classifier, leading to unfair comparison.
> Questions:
> More fine-grained ablation experiments and comparisons are helpful: 1. How about only using the last or last two layers to extract features in SV? 2. How about sharing the learnable parameter for each category in HV? 3. How about the comparisons of FLOPs and the number of parameters?

---

> ### Author Rebuttal · Authors · 2024-08-07
>
> Thanks for your constructive comment. Here are the responses to Weaknesses (W), Questions (Q) and Limitations(L).
>
> **W1. The design of the main modules lacks novelty and in-depth exploration**
>
> **Novelty.**
> We fully agree with your summary of the modules within our model.
> However, the crucial aspect is that we construct a framework capable of leveraging intratumoral tissue heterogeneity
> and utilizing node types to introduce pathology priors into the model.
> To our knowledge, this is novel in the computational pathology literature.
>
> We also introduced a more suitable prototype extraction method tailored for computational pathology tasks.
> In existing prototype-based computational pathology networks,
> prototypes are often extracted simply based on fixed cluster centers,
> which may struggle to capture the full variability or complexity of the features.
> By contrast, in our approach,
> We design learnable shifts to extend each prior-based prototype into multi-prototypes with different preferences.
> These prototypes focus on interactions with different tissues and learn from previously unknown factors or phenotypes that may contribute to patient outcomes.
>
> Additionally, we provide an interpretability figure illustrating the interactions between prototypes from different categories and the global tissue,
> which allows for a better exploration of the global tissue interactions discovered by the model.
> We have expanded the interpretability figure in the PDF attachment in "Author Rebuttal",
> if you are interested about it, please refer to "Author Rebuttal" for more details.
>
> **In-depth exploration.**
> For the finer-grained ablation experiments and comparisons you suggested,
> we add ablation studies in the following sections.
> If you believe there are other areas that require in-depth exploration, we are willing to include additional experiments.
>
> **W2. ProtoSurv requires additional labels to train a classifier, leading to unfair comparison**
>
> We used classifier-obtained node types to inject pathology priors into the model,
> aiming to explore the benefits of incorporating domain priors.
> The experimental results indicate that the model indeed improved with the assistance of these domain priors.
> In the ablation experiments shown in Table 4 of the main paper,
> we demonstrated that when using publicly available classifiers or simply using K-means to obtain node categories,
> our framework still achieved better results than the compared models.
> This indicates that in the fair comparison without the advantage of pretrained classifier node types,
> our framework's ability to handle tumor heterogeneity still leads to superior prediction performance compared to baselines.
>
> **Q1. How about only using the last or last two layers to extract features in SV?**
> Thanks for the advices.
> We tested using only the last and the last two layers to extract features in SV.
> The experimental results are as follows:
>
> ||COAD|LGG|LUAD|PAAD|BRCA|  average  |
> |:-:|:-:|:-:|:-:|:-:|:-:|:-:|
> | ProtoSurv (SV all layers, n=4) |   0.692 ± 0.045| **0.774 ± 0.063** | **0.658 ± 0.046**|0.669 ± 0.049|0.720 ± 0.040| **0.703** |
> |   ProtoSurv (SV last layer)|   0.678 ± 0.051|   0.764 ± 0.054   | **0.658 ± 0.060**|**0.671 ± 0.042** |0.718 ± 0.049|0.698|
> | ProtoSurv (SV last two layers) | **0.693 ± 0.057** |0.762 ± 0.037|0.656 ± 0.058|0.662 ± 0.049|**0.723 ± 0.044** |0.699|
>
> Although the optimal results varied across different datasets, overall,
> models that used more layers to extract features achieved better results.
>
> **Q2. How about sharing the learnable parameter for each category in HV?**
>
> We greatly appreciate your advice.
> Sharing the learnable parameters for each category can significantly reduce the model's parameter.
> This helps explore the scalability potential of the model.
> Here are the results of sharing the learnable parameters for each category.
>
> ||COAD|LGG|LUAD|PAAD|BRCA|average|
> |:-:|:-:|:-:|:-:|:-:|:-:|:-:|
> |ProtoSurv| **0.692 ± 0.045** | **0.774 ± 0.063** | **0.658 ± 0.046** |**0.669 ± 0.049** |**0.720 ± 0.040**| **0.703** |
> |ProtoSurv(Sharing parameters)|0.667 ± 0.055|0.765 ± 0.044|0.653 ± 0.042|0.652 ± 0.053|0.707 ± 0.026|0.689|
>
> We show the optimization of computational requirements brought by sharing the learnable parameters in Q3.
>
> **Q3. How about the comparisons of FLOPs and the number of parameters?**
>
> Follow your advice,
> we evaluate the model's inference time, floating point of operations(FLOPs), model parameters, and maximum GPU memory usage.
> We use a WSI which contains 32,625 patches as input.
> The computation time is measured using a Nvidia RTX 3090 GPU.
>
> We included PatchGCN for comparison.
> We additionally test ProtoSurv-tiny under a reduced parameter configuration (prototype dim = 256, hidden dim of SV and HV = 64, prototypes per category = 4),
> to evaluate the performance degradation of our architecture with fewer parameters and its scalability for more limited hardware.
>
> || PatchGCN | ProtoSurv | ProtoSurv-tiny | ProtoSurv(Sharing parameters) |
> |:-:|:-:|:-:|:-:|:-:|
> |Time(s)|0.12|0.29|0.21|0.29|
> |FLOPs(G)|  30.49|627.3|96.5|627.3|
> |Number of Parameters(M)|1.19|39.1|4.77|15.5|
> |Maximum GPU memory usage(MB) |1570|5417|1523|5326|
>
> Here are the survival prediction results of ProtoSurv under the reduced parameter configuration.
>
> ||COAD|LGG|LUAD|PAAD|BRCA|
> |:-:|:-:|:-:|:-:|:-:|:-:|
> |Patch-GCN|0.652 ± 0.086|0.713 ± 0.054|0.635 ± 0.027|0.618 ± 0.057   |0.647 ± 0.032|
> |ProtoSurv|**0.692 ± 0.045** |**0.774 ± 0.063** |0.658 ± 0.046|0.669 ± 0.049| **0.720 ± 0.040**|
> |ProtoSurv-tiny|0.673 ± 0.039|0.756 ± 0.038| **0.664 ± 0.039** | **0.687 ± 0.049**|0.707 ± 0.044|
> | ProtoSurv(Sharing parameters) |0.667 ± 0.055|0.765 ± 0.044|0.653 ± 0.042|0.652 ± 0.053|0.707 ± 0.026|
>
> Once again, thank you for your careful review and valuable comments on our paper.
> If you are satisfied with our response,
> would you kindly bump up your score?

---

> > ### Comment · Reviewer_gRNp · 2024-08-13
> >
> > thanks for the detailed reply from the authors, which addressed most of my concerns.
> > According to the opinion of other reviewer, this paper is at a marginal level, and I am inclined to accept it at the marginal level.

---

### Official Review · Reviewer_tNQM · 2024-07-12

**Soundness:** 2
**Presentation:** 2
**Contribution:** 2
**Rating:** 5
**Confidence:** 4

**Summary:**

The authors proposed ProtoSurv, a graph model for WSI survival prediction. The key contribution is learning different prototypes for each node type, and aggregating nodes using cross attention and learned prototypes.

**Strengths:**

- Outcome prediction for cancer patients is a very relevant and important problem and has a large impact. Recently, there has been a lot of interest in the field of computational pathology to solve this problem.
- ProtoSurv has a good motivation to identify and aggregate based on prototypes, which provides some inherent interpretability to the model.
- ProtoSurv seems to outperform various baselines reported by the authors.

**Weaknesses:**

- Having a fixed set of hand crafted node types and a fixed number of multi-prototypes is limiting, and restricts the model’s ability to learn from previously unknown factors / phenotypes that may contribute to patient outcome.
- Evaluation benchmarks should include popular aggregation baselines, including ABMIL (used by UNI authors for benchmarking) and TransMIL.
- Ablation studies is not complete. The authors should consider benchmarking:
  - Use UNI features directly for prototype learning, without considering node types;
  - Removing graph network, or replacing it with a transformer.

**Questions:**

Please address the points in the weakness section.

**Limitations:**

Limitations are addressed in the paper.

---

> ### Author Rebuttal · Authors · 2024-08-07
>
> Thanks for your constructive comment. Here are the responses to Weaknesses (W).
>
> **W1. Having a fixed set of hand crafted node types and a fixed number of multi-prototypes is limiting,
> and restricts the model’s ability to learn from previously unknown factors / phenotypes that may contribute to patient outcome.**
>
> We fully agree with your comment.
> We consider that simply extracting prototypes for each node category based on priors
> might indeed restrict the model's ability to learn from previously unknown factors / phenotypes that may contribute to patient outcome.
>
> Therefore, when designing the prototype extraction module (the Histology View (HV) module),
> We made the following efforts to ensure the model is not constrained by node types,
> and to enhance the model's ability to learn from factors/phenotypes beyond the priors:
>
> 1. We rely solely on node types provided by prior knowledge to give each prototype an initial preference,
> rather than using them as constraints for the prototypes.
>
> 2. Then we use learnable shifts to obtain diverse prototypes for each category,
> enabling the model to prefer on different factors within the category,
> including both previously known and unknown factors / phenotypes.
>
> 3. Finally, under the guidance of these preferences,
> the prototypes to aggregate relevant information from the global features by learnable cross-attention.
>
> In summary, our prototype extraction process is learnable and highly flexible.
> The node types only provide an initial preference for each prototype, rather than a constraint.
> The fixed number of multiple prototypes obtained through offsets increases the diversity of prototypes within each category,
> encouraging the model to have more varied preferences. Finally, through learnable cross-attention,
> each prototype extracts relevant features from the global context (without being limited by node types),
> further enhancing the model’s ability to learn from previously unknown factors/phenotypes that may contribute to patient outcomes.
>
> We apologize for any misunderstanding caused by the description in our paper.
> We will revise this section in the camera-ready version to more clearly explain our prototype extraction process.
> Additionally, we have uploaded a detailed interpretability figure in the PDF attachment in "Author Rebuttal",
> which visualizes the attention preferences of multiple prototypes across different categories to further support our claims.
> If you are interested about it, please refer to "Author Rebuttal" for more details.
>
> **W2. Evaluation benchmarks should include popular aggregation baselines, including ABMIL (used by UNI authors for benchmarking) and TransMIL.**
>
> We include additional experiments with popular aggregation baselines, the results are as follows:
>
> |           |       COAD        |        LGG        |       LUAD        |       PAAD        |       BRCA        |
> |:---------:|:-----------------:|:-----------------:|:-----------------:|:-----------------:|:-----------------:|
> |   ABMIL   |   0.647 ± 0.036   |   0.710 ± 0.048   |   0.653 ± 0.059   |   0.625 ± 0.063   |   0.657 ± 0.064   |
> | TransMIL  | **0.695 ± 0.051** |   0.739 ± 0.034   |   0.608 ± 0.040   |   0.642 ± 0.037   |   0.694 ± 0.053   |
> | ProtoSurv |   0.692 ± 0.045   | **0.774 ± 0.063** | **0.658 ± 0.046** | **0.669 ± 0.049** | **0.720 ± 0.040** |
>
> **W3(1). Ablation study: Use UNI features directly for prototype learning, without considering node types.**
>
> Following your advice, we supplement the ablation experiments here.
> In this ablation experiment,
> we used 8 cluster centers obtained from the K-means algorithm as the initial values of each prototype (corresponding to the process of eq (4) in the main paper),
> and performed prototype learning directly without considering node types.
> Here are the experimental results:
>
> |                                        |       COAD        |        LGG        |       LUAD        |       PAAD        |       BRCA        |
> |:--------------------------------------:|:-----------------:|:-----------------:|:-----------------:|:-----------------:|:-----------------:|
> |               ProtoSurv                | **0.692 ± 0.045** | **0.774 ± 0.063** | **0.658 ± 0.046** |   0.669 ± 0.049   | **0.720 ± 0.040** |
> | ProtoSurv(K-means cluster center) |   0.674 ± 0.044   |   0.763 ± 0.065   | **0.658 ± 0.054** | **0.671 ± 0.054** |   0.718 ± 0.041   |
>
> **W3(2). Ablation study: Removing graph network, or replacing it with a transformer.**
>
> We performed an ablation experiment on removing the graph network in the "without (w/o) SV" row of Table 2 in the main paper.
> However, we acknowledge that this experiment is still insufficient.
> Following your advice, we supplement the ablation experiments here.
> We remove the graph network (Structure View module (SV)) while retaining the prototype extraction module (Histology View module (HV)),
> and test the performance of the prototypes extracted by HV under different pooling methods.
> We test mean pooling by calculates the average value of all prototypes (HV(mean pooling) row),
> and concat pooling by concatenating all prototypes along the channel dimension(HV(concat pooling) row).
>
> Here are the experimental results:
>
> |                    |       COAD        |        LGG        |       LUAD        |       PAAD        |       BRCA        |
> |:------------------:|:-----------------:|:-----------------:|:-----------------:|:-----------------:|:-----------------:|
> |     ProtoSurv      | **0.692 ± 0.045** | **0.774 ± 0.063** | **0.658 ± 0.046** | **0.669 ± 0.049** | **0.720 ± 0.040** |
> |  HV(mean pooling)  |   0.684 ± 0.044   |   0.706 ± 0.036   |   0.646 ± 0.051   |   0.624 ± 0.032   |   0.657 ± 0.049   |
> | HV(concat pooling) |   0.688 ± 0.046   |   0.766 ± 0.044   |   0.641 ± 0.037   |   0.661 ± 0.057   |   0.713 ± 0.039   |
>
>
> Once again, thank you for your careful review and valuable comments on our paper.
> If you are satisfied with our response,
> would you kindly bump up your score?

---

### Official Review · Reviewer_DhDE · 2024-07-13

**Soundness:** 2
**Presentation:** 3
**Contribution:** 2
**Rating:** 5
**Confidence:** 4

**Summary:**

This paper introduces ProtoSurv, an algorithm which performs survival prediction for Whole Slide Images (WSIs) of tumour samples, by taking into account the interaction between different tissue types and tumour heterogeneity. ProtoSurv proposes leveraging prior tissue knowledge by constructing a graph where the nodes represent patches and edges represent spatial relationship. The attributes of the nodes correspond to embedded feature representation of the patch, as well as a tissue category label. This tissue category label is assigned by a finetuned feature extractor (UNI), which was previously modified by adding a classifier head and training it on patches with know tissue type labels. Once the graph is initialised, the pipeline employs a dual-view architecture, composed of a Structure view and a Histology view. In the Structure view, the graph with its embedded feature representation is passed through a GCN (which correspond to the Patch-GCN architecture) and a final representation H is stored. In the Histology view multiple prototypes are learned for each tissue category, aiming to capture tissue heterogeneity by using cross-attention between the initial feature vector and the prototypes. Cross-attention is then also employed to guide the fusion of the information from the Structure and the Histology view. The loss function employed during training correspond to composition of the Cox regression loss + a modified compatibility loss + an orthogonality loss. The pipeline is applied to 5 TCGA cancer datasets and compared to several SOTA baseline models. The results are presented using C-index. There is ablation on different model components, the number of tissue categories, the feature extractor used and number of prototypes. The results provided show the algorithm outperforms against the baseline models.

**Strengths:**

This paper presents an interesting dual-stream architecture for cancer survival prediction, combining a spatial graph based approach, with a prototype learning module. Overall the paper is well structured, and the incorporation of prior tissue knowledge to guide cross-attention is a nice contribution which grounds the model in clinical understanding. The method is tested comprehensively on 5 benchmark methods and ablation is provided on the different components of the model. The results obtained show improvement in Survival prediction across the datasets.

**Weaknesses:**

Below I list what I perceive to be the main weaknesses in this paper:

1 - This is not a heterogeneous graph neural network: the edges in the graph are created based on spatial proximity of patches, regardless of their tissue type, and while each node has two types of attributes (Feature vector and Tissue category label) the tissue label isn't actually used in the GNN, which operates on a standard homogeneous graph structure. I find this problematic because its a significant misrepresentation of the core methodology in my opinion. Instead, this is a dual stream architecture that processes the data in two different ways (Patch-GCN + prototype learning), before combining them using cross-attention. Given this fact it would be good to compare how this method compares to a simple ensemble of Patch-GCN and a prototype-based model to verify model performance against this baseline.

2 - As this pipeline relies heavily on cross-attention, both in the histology view and in the fusion module, I think the authors should discuss computational requirements and scalability of the model to large WSIs, which is an important consideration in the healthcare setting.

3 - The ablation on the PGF module is unclear: it seems the PGF module is replaced with an aggregation method from Dong et al., where the order of Q and KV in the cross-attention mechanism is swapped. This doesn't completely remove the fusion between the two views, rather it changes how the fusion is performed and hence does not provide a clear understanding of the individual contribution of the PGF module. Maybe adding a comparison to a simple concatenation and classification could add clarity to this point.

4 - The authors claim improved interpretability due to the use of prototypes and they have an interesting Figure 10 in the Appendix, which I think should be referenced in text. It would be good to clarify the model obtains attention maps per category and per prototype, and if possible include the Figure in the main body of the text as this would back the improved interpretability claim.

5 - One important comment is that the structural view doesn't simulate viewing at multiple magnification or scales as stated in the text, rather it merely extends the receptive field. This provides multi-hop neighbourhood information, but not multi-scale information.

**Questions:**

I have addressed my questions in the Weaknesses section. Below are some general comments:

- Line 1 - "Tumors are"
- Line 18 - slide
- Line 21 - molecular alterations
- Line 22 - no need to redefine WSIs acronym
- Line 24 - you could cite Ilse et al. 2018 here.
- Line 26 - In MIL aggregation to bag-level representation can be done using non-learnable or learnable aggregation layers. MIL methods which use learnable aggregation do learn about interrelationship among instances, depending on the approach employed.
- Line 29 - I don't agree with this statement, subtyping and staging can absolutely require a holistic view of the tumour micro-environment.
- Line 47 - I would qualify this statement: "it could be affecting" - there is a lot of research and debate the performance of GNNs on homophilous vs heterophilous graph. For example see [1].
- Line 56 - 59 - I don't find these sentences very clear, maybe you could reformulate them?
- Line 75 - I imagine most approaches using CNNs would also be employing a MIL based approach.
- Line 80 + 87 - "context-awareness"
- Line 93 - This was done before HEAT, see [2]
- Line 106 - See also [3]
- Line 114 - $e_{i,j}$
- Line 115 - "where $x_i$ is the feature ..." ?
- Line 130 - " by the current consensus as* highly relevant"
- Line 213 - "We employ*"
- Line 225 - "given the morphological alterations found in frozen sections*"
- Table 1 - underlined*
- Line 310 - "In our model, we incorporate five tissue categories based on pathological knowledge of prognosis-related tissues."

1 - [1] Platonov etal., A critical look at evaluation of GNNs under heterophily: Are we really making progress, ICLR, 2023.

2 -  [2] Pati et al., Hierarchical graph representations in digital pathology, MIA, 2022.

3 - [3] Yu et al., Prototypical multiple instance learning for predicting lymph node metastasis of breast cancer from whole-slide pathological images, MIA, 2023.

**Limitations:**

The authors mention an important limitation, which is that the tissue categories are fixed and obtaining the labels remains an obstacle. I would expand on this and mention the method relies on a pre-trained tissue classifier, but there's limited discussion on how errors in this classification might impact the overall performance. I also think the heavy use of cross attention mechanism used could introduce problem for scaling to a clinical setting.

---

> ### Author Rebuttal · Authors · 2024-08-07
>
> Thanks for your constructive comment. Here are the responses to Weaknesses (W), Questions (Q) and Limitations(L).
> Due to word limit, we omit many details.
> If you have any further questions, we would be willing to response.
>
> **W1. This is not a heterogeneous graph neural network.**
>
> We describe our model as a heterogeneous graph neural network in the paper,
> because we were indeed inspired by the optimization of heterogeneous graphs of aggregating information from global nodes [1].
> Similar to them, each node in our model has a category which participates in guiding the global feature extraction.
> Therefore, we cautiously feel that it is appropriate to define the model as a heterogeneous graph-based model.
>
> However, we believe that the dual-stream architecture indeed better captures the model's characteristics.
> Each module has its own focus and does not need to be collectively described as a heterogeneous graph.
> We will revise the description to highlight the characteristics of its dual-stream architecture.
>
> [1] Li et al. Finding global homophily in graph neural networks when meeting heterophily, ICML2022.
>
> **W1(1). Compare with Patch-GCN and prototype-based models.**
>
> Here, we provide a more detailed ablation experiments about prototypes.
> Additionally, as a complement comparison to the prototype-based model,
> we reference PANTHER [2], a SOTA prototype-based unsupervised model, for comparison.
> We test mean pooling and concat pooling of Histology View (HV).
> Same as PANTHER, concat pooling concatenate all prototypes along the feature dimension.
>
> Here are the results:
>
> ||COAD|LGG|LUAD|PAAD|BRCA|
> |:-:|:-:|:-:|:-:|:-:|:-:|
> |Patch-GCN|   0.652 ± 0.086|0.713 ± 0.054|0.635 ± 0.027|0.618 ± 0.057|0.647 ± 0.032   |
> |ProtoSurv| **0.692 ± 0.045** | **0.774 ± 0.063** | **0.658 ± 0.046** |0.669 ± 0.049| **0.720 ± 0.040** |
> |PANTHER|   0.635 ± 0.056   |0.748 ± 0.046|0.631 ± 0.029   | **0.673 ± 0.082** |0.699 ± 0.019   |
> |  HV(mean pooling)  |0.684 ± 0.044|0.706 ± 0.036|0.646 ± 0.051|0.624 ± 0.032|0.657 ± 0.049   |
> | HV(concat pooling) |0.688 ± 0.046|0.766 ± 0.044|0.641 ± 0.037|0.661 ± 0.057|0.713 ± 0.039   |
>
> [2] Song et al. Morphological prototyping for unsupervised slide representation learning in computational pathology, CVPR2024.
>
> **W2. Computational requirements and scalability**
>
> **Computational requirements.**
> We evaluate the model's inference time, floating point of operations(FLOPs), model parameters, and maximum GPU memory usage.
> We use a WSI which contains 32,625 patches as input.
> The computation time is measured using a Nvidia RTX 3090 GPU.
> We additionally test ProtoSurv-tiny under a reduced parameter configuration
> (prototype dim = 256, hidden dim of SV and HV = 64, prototypes per category = 4).
>
> || ProtoSurv | ProtoSurv-tiny | PatchGCN |
> |:-:|:-:|:-:|:-:|
> |Time(s)|   0.29|0.21|   0.12   |
> |FLOPs(G)|   627.3|96.5|  30.49|
> |Model Parameters(M)|39.1|4.77|1.19|
> | Maximum GPU memory usage(MB) |5417|1523|1570|
>
> Here are the results of ProtoSurv-tiny.
>
> ||COAD|LGG|LUAD|PAAD|BRCA|
> |:-:|:-:|:-:|:-:|:-:|:-:|
> |   ProtoSurv    | **0.692 ± 0.045** | **0.774 ± 0.063** |0.658 ± 0.046|0.669 ± 0.049   | **0.720 ± 0.040** |
> | ProtoSurv-tiny |0.673 ± 0.039|0.756 ± 0.038| **0.664 ± 0.039** | **0.687 ± 0.049** |0.707 ± 0.044   |
>
> **Scalability.**
> The HV and SV modules, as well as the process of extracting prototypes, are completely decoupled,
> and can be computed in parallel.
>
> **W3. The ablation on the PGF module.**
>
> Here, we include a supplement ablation study for the PGF module.
> In this ablation study,
> we remove the fusion between the two views and instead concatenated directly along the patch dimension.
>
> ||COAD|LGG|LUAD|PAAD|BRCA|
> |:-:|:-:|:-:|:-:|:-:|:-:|
> |ProtoSurv| **0.692 ± 0.045**| **0.774 ± 0.063** |0.658 ± 0.046| **0.669 ± 0.049** | **0.720 ± 0.040** |
> |w/o PGF(concat) |0.659 ± 0.058|0.712 ± 0.084| **0.662 ± 0.064**|0.652 ± 0.040|0.719 ± 0.024   |
>
>
> **W4. An interesting Figure 10 in the Appendix.**
>
> We greatly appreciate your interest of Figure 10.
> As per your suggestion,
> we will reorganize the camera-ready version to move it to "Experiments" section.
> In addition, we have expanded the figure,
> please refer to the PDF attachment in "Author Rebuttal" for more details.
>
> **W5. SV doesn't simulate viewing at multiple magnification.**
>
> GNNs extend the receptive field, and each GNN layer provides different neighborhood information.
> In SV, all output from each GNN layer are concatenated,
> which aggregate features with varying hops of neighborhood and different receptive fields.
> We full agree with you that it strictly provides multi-hop neighborhood information rather than multi-scale information.
> We will revise the description to make it more precise.
>
> **L1. Discussion on how errors in classification might impact the overall performance.**
>
> To minimize the impact of classification errors on the overall performance,
> in the HV module, we rely only on the node categories provided by the classifier to delineate an initial range.
> We calculate the average value of the patch features within the category range as the initial prototype,
> providing an initial preference for the aggregation of each prototype (pathology prior injection).
>
> In the main paper's Table 4,
> we demonstrated the robustness of our model using ablation results with publicly available classifiers and clustering methods.
> To further illustrate this point, we randomly generated categories for 20% and 30% of the nodes.
> The experimental results are as follows:
>
> ||COAD|LGG|LUAD|PAAD|BRCA|
> |:-:|:-:|:-:|:-:|:-:|:-:|
> |ProtoSurv  | **0.692 ± 0.045** |0.774 ± 0.063| **0.658 ± 0.046** |0.669 ± 0.049| **0.720 ± 0.040** |
> |20% random|0.685 ± 0.051|0.769 ± 0.048| **0.658 ± 0.046** | **0.671 ± 0.044** |0.712 ± 0.045   |
> |30% random|0.689 ± 0.053| **0.777 ± 0.047** |0.656 ± 0.043|0.666 ± 0.047|0.717 ± 0.047   |
>
> Once again, thank you for your careful review and valuable comments.
> If you are satisfied with our response,
> would you kindly bump up your score?

---

> > ### Comment · Reviewer_DhDE · 2024-08-12
> > **Response to Author's Rebuttal**
> >
> > I thank thank the authors for their comprehensive rebuttal. In light of their answers I have updated my rating.

---

### Author Rebuttal · Authors · 2024-08-07

We thank all reviewers for the valuable feedbacks and constructive comments.

We have responded to each reviewer's comments.

We attach a PDF containing an expanded interpretability figure.
From the interpretability figure,
we observed that the attention preferences of multi-prototypes from a category varied.
Some prototypes were responsible for extracting global category information,
while others focused on discovering interactions between other categories.
This indicates that our prototype learning paradigm has the potential to uncover unknown interactions and factors.

---

### Decision · Program_Chairs · 2024-09-25

**Decision:**

Accept (poster)

**Comment:**

The paper introduces ProtoSurv a graph neural network to encode tissue structures in whole slide images by embedding patches using tuned UNI model and considering them as nodes of graph and edges are considered to encode tissue spatial relationships. The idea is interesting which makes the interaction between local and regional tissue features in WSI to be explored simultaneously. This is done via three stages of operations i.e. SPBD, SPRS and HGT. The proposed pipeline outputs two different views of WSI i.e. structure view and histology to capture tissue heterogeneity and predict cancer survival prediction. The paper has received all marginal accept reviews. There is consensus among reviewers that the utility of ProtoSurv is incrementally novel where the incorporation of prior tissue knowledge grounds the model for clinical understanding. The paper is well structured and written. Comprehensive evaluations are done on five TCGA benchmark datasets and compared with several baseline methods where proposed method demonstrates its utility to enhance survival prediction accuracy. Some concerns were raised by reviewers on (a) validity of heterogenous graph neural network, (b) discussion on computational requirements and scalability of the model to large WSI (e.g. surgical slides), (c) unclear and insufficient ablation studies, (d) interpretability of the pipeline, (e) ability to simulate structural view at multiple magnification levels, (f) comparison with MIL aggregation baselines, (g) over statement of claims regarding overfitting via SPBD, (h) introduction of multiple hyper-parameters in the proposed method and how they influence different stages of the proposed algorithm, and (i) preselection of prototypes prior to training which can undermine its adaptation/generalization. Authors have addressed several concerns during rebuttal and provided further comparison and ablation studies. The AC finds enough support from the reviewers to merit the paper for publication. It is highly encouraged for the authors to take the advantage on the discussions raised by reviewers as highlighted above for their final revision using from both pre-/post-rebuttal phase comments.